**Data Availability Statement:** All relevant data are within the manuscript and its Supporting Information files.

**Funding:** [This research was supported by National Natural Science Foundation of China (Nos.

# Construction project risk prediction model based on EW-FAHP and one dimensional convolution neural network

Yawen Zhong[1]*, Hailing Li[2], Leilei Chen[3]

**1** School of Engineering, Southwest Petroleum University, Nanchong, China, **2** School of Civil Engineering and Environment, Xihua University, Chengdu, China, **3** Sichuan University of Science and Engineering, the Artificial Intelligence Key Laboratory of Sichuan Province, Zigong, China

* arwenzyw@163.com

## Abstract

In order to solve the problem of low accuracy of traditional construction project risk prediction, a project risk prediction model based on EW-FAHP and 1D-CNN(One Dimensional Convolution Neural Network) is proposed. Firstly, the risk evaluation index value of construction project is selected by literature analysis method, and the comprehensive weight of risk index is obtained by combining entropy weight method (EW) and fuzzy analytic hierarchy process (FAHP). The risk weight is input into the 1D-CNN model for training and learning, and the prediction values of construction period risk and cost risk are output to realize the risk prediction. The experimental results show that the average absolute error of the construction period risk and cost risk of the risk prediction model proposed in this paper is below 0.1%, which can meet the risk prediction of construction projects with high accuracy.

## Introduction

With the continuous development of science and technology, the complexity of construction projects continues to increase, the construction period continues to grow, and there are many uncertain factors [1], in order to reduce the probability of risk occurrence and effectively avoid potential risks to the entire project, it is necessary to predict the risks of the construction projects.

In reference [2], the subway project construction risk management method is based on Bayesian network. In reference [3], the Fault Tree Analysis (FTA) method is combined with Bayesian network, and a Bayesian network-based shale gas well blowout risk analysis method is proposed. However, Bayesian networks are based on prior probabilities. In many cases, prior probabilities depend on assumptions, which will largely lead to poor prediction results. Combine with AHP theory, use rough set to analyze project risk group decision to realize attribute reduction, and combine with analytic hierarchy process to realize quantitative research and analysis of project risk. Literature [4] uses AHP to evaluate solid waste treatment methods in Libya, but it is difficult for AHP to check and adjust the consistency of the judgment matrix. Literature [5] proposed the use of fuzzy analytic hierarchy process and rough analytic

11705122, 61640223), Sichuan Provincial Department of Science and Technology Project (No. 2019YJ0477), Artificial intelligence Sichuan Key Laboratory Project (No.2019RYY01), Nanchong Science and Technology Bureau Project (No.19SXHZ0040)]

**Competing interests:** The authors have declared that no competing interests exist.

hierarchy process to evaluate traffic accessibility method. Literature [6] proposed the use of IT2FS-DEMATEL to eliminate less important indicators, combines the IT2FS-AHP method to sort the final indicators, and establishes a multi-index decision-making model. But literature [5] and literature [6] involve risk prediction, rough set and IT2FS-DEMATEL may have a greater impact on the final prediction results after removing redundant attributes.

With the development of artificial intelligence and big data, neural network has attracted more and more researchers' attention. Because neural network has a strong nonlinear fitting ability and has a good effect on mapping nonlinear relations, relevant scholars have combined neural network with engineering project risk research in recent years and achieved certain results [6,7]. Literature [8] proposed a railway construction risk assessment algorithm based on BP neural network. The expert scoring method was used to establish initial sample data, and the BP neural network prediction model was used to learn and predict the samples to get the risk score of each construction project. However, BP neural network has some problems, such as not the best approximation of continuous function and long training time. Literature [9] proposed a project risk evaluation algorithm based on PCA (principal component analysis)-RBF neural network on the basis of BP model, which improved the shortcomings of BP neural network that it is difficult to obtain the optimal network, but the RBF neural network the center of the hidden basis function is selected in the input sample set, which in many cases can hardly reflect the real input-output relationship of the system. In order to solve the problems of the above-mentioned neural network, literature [10] proposed a method for predicting the risk of underground engineering rockburst based on ANN and ABC. (the artificial neural network (ANN) and artificial bee colony (ABC), in order to further improve the prediction accuracy, the paper uses the artificial bee colony algorithm to optimize the artificial bee colony algorithm, but the artificial bee colony algorithm has weak search ability and relatively slow convergence speed.

As construction projects become large-scale and risk factors continue to increase, traditional risk predictions mostly use regular event analysis, correlation analysis and other methods to analyze key indicators and detailed records, which rely heavily on manual extraction by professional workers. The current risk assessment of construction projects adopts a single expert scoring method, entropy weight method, analytic hierarchy process or fuzzy analytic hierarchy process, which does not fully combine multiple evaluation methods, resulting in incomplete detailed factors affecting project risks, and lack of objectivity and accuracy of inspection and evaluation results. At present, Analytic Hierarchy Process (AHP) and its derivative methods are still the most widely used and most effective risk assessment in the complex large systems. Among them, the Fuzzy Analytic Hierarchy Process (FAHP), which integrates fuzzy theories, improves the weight determination problem of AHP, and its practicality and simplicity have been applied more and more widely [11]. In order to improve the closeness between the weight of evaluation index and reality, this paper adopts the entropy weight-fuzzy analytic hierarchy process (EW-FAHP method) to determine the weight. The risk prediction method based on traditional neural network risk prediction requires too many weights, which reduces the accuracy of project prediction to a certain extent [12]. The current risk assessment of construction projects uses a single CNN network with different convolution kernels to perform convolution operations on the input data, thereby obtaining global features of the data, and then down-sampling the extracted features through pooling operations, reducing the amount of calculations. At the same time, it can also suppress the overfitting of the model to a certain extent.

Therefore, this paper uses entropy weight method and fuzzy analytic hierarchy process to evaluate the construction period and cost index system of the construction project, proposes a construction project risk prediction model based on EW-FAHP and 1D-CNN, identifies the existing risks of the construction project through reference analytical method and constructs risk evaluation index system. Using Entropy Weight (EW) and Fuzzy Analytic Hierarchy

Process (FAHP), the risk weight of each risk evaluation index is determined by combining subjective and objective evaluation methods. One dimensional convolution neural network model is constructed to train and learn the risk weight of construction project. The duration risk and cost risk of construction project are selected as the output unit of convolution neural network. The average absolute error between the predicted value and the actual value of duration risk and cost risk is analyzed to realize the risk prediction of construction project.

The prediction results of the risk prediction model proposed in this paper show that the method has strong practicability in the early stage of the project and in the construction process. Compared with other commonly used forecasting algorithms, the forecasting accuracy has been significantly improved, which is of greater reference value for project decision-makers.

## Project risk theory research

### Project risk identification

Starting from the decision-making stage, various risks affecting the construction project duration and costs will arise as the project progresses. For the construction project management and construction parties, it is necessary to target the entire implementation process of the project with limited resources. Accurately identify the risk factors that have a greater impact on the construction period and costs [13–15]. For construction project management and construction parties, it is necessary to accurately identify the risk factors that have a greater impact on the construction period and cost for the entire implementation process of the project under the condition of limited resources.

Project risk identification methods mainly include: brainstorming method, literature research method and rough set theory. Each method has its best applicable environment, and suitable identification methods can be selected according to different analysis angles, routes and focuses [16]. Compared with other identification methods, literature research method is not limited by time and space, and can realize risk identification even with a small amount of resources. This method has been widely applied in intelligent algorithms, big data analysis, fault diagnosis, etc. [17,18].

This paper selects construction projects invested by state-owned assets, controlled by state-owned assets or directly managed by government departments for analysis. Therefore, literature analysis is adopted to identify risk factors and summarize project risk evaluation indexes. Combined with the attributes of construction project risk, the project risk preliminary evaluation index system is obtained.

### Project risk assessment

In project management, project risk evaluation refers to the process of analyzing, estimating and quantifying the impact of risks on the project. Establishing a scientific and effective risk assessment method is the prerequisite for risk research. The flowchart of the risk assessment process is shown in Fig 1.

There are many methods for quantitative analysis of risk, because there are some risks in large construction projects that cannot be evaluated by statistical and mathematical modeling methods. In order to improve the consistency of analytic hierarchy process(AHP), this paper selects expert scoring method and fuzzy comprehensive evaluation method to evaluate the risk factors.

### Construction of project risk evaluation indicators

In construction project management, choosing an appropriate risk evaluation index system is the prerequisite for controlling project risks. The selection of risk assessment indicators should meet the requirements of representativeness, diversity, conciseness and comprehensiveness

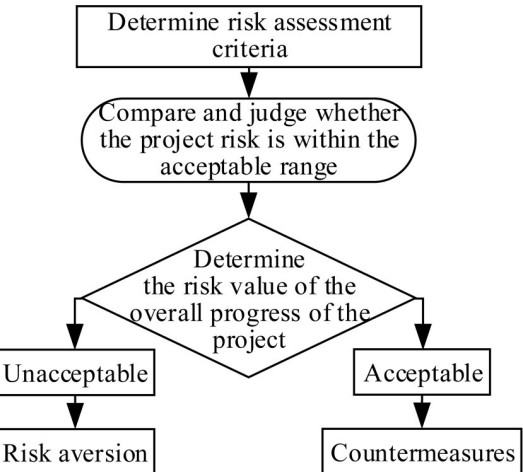

**Fig 1. Flow chart of risk assessment process.**

[19,20]. The hierarchy of the system determines whether the evaluation index system is scientific and reasonable. Therefore, when constructing the project risk evaluation index system, the evaluation indexes are divided according to the defined grade categories, and finally a multi-level index system is constructed to help risk managers understand the specific conditions of the risks in the project more comprehensively [21,22]. As is shown in Fig 2, the primary indicators are $R_i(i = 1,2,L\ 6)$, and the secondary indicators corresponding to each primary indicator are $U_i(i = 1,2,L\ 25)$.

## Risk assessment of construction project

The evaluation of risk is measured by the degree of deviation between the final result of the project and the previous target. The degree of deviation is positively correlated with the risk. The greater the degree of deviation, the greater the risk.

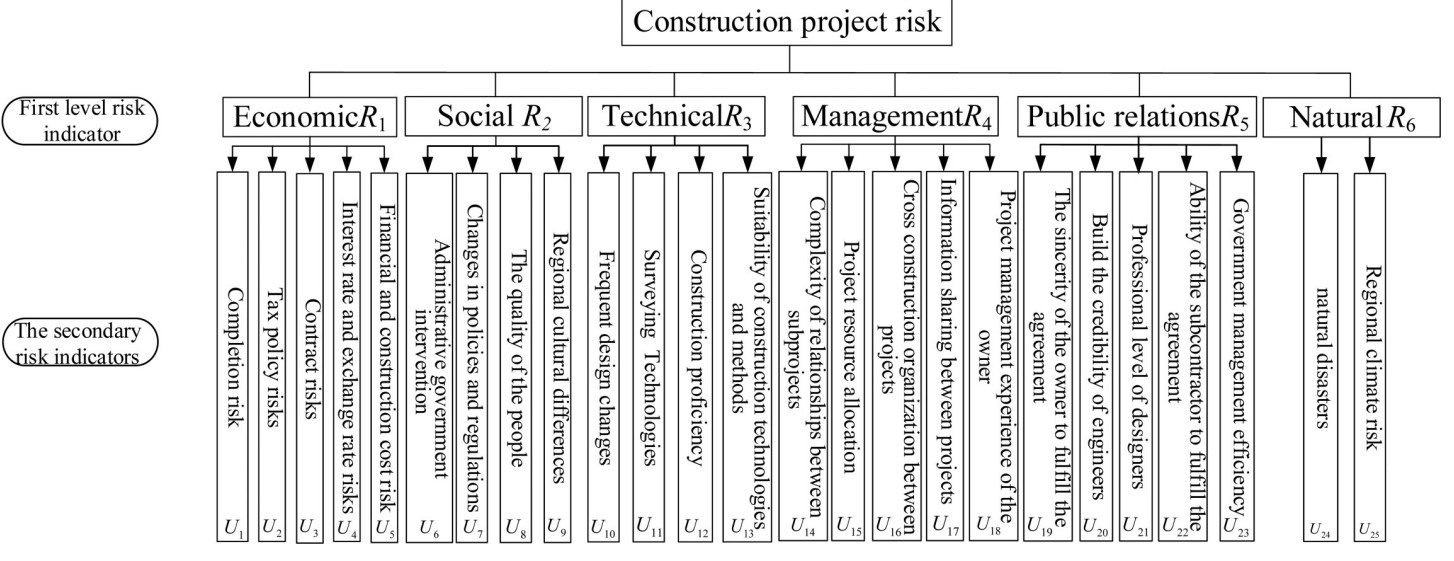

**Fig 2. Evaluation index system.**

Through the research and analysis of interval fuzzy numbers, it is found that the existing processing method is to directly model and predict the two boundary points. Doing so often leads to a failure to describe the overall development trend of the sequence and the results predicted by the model are prone to be confused, etc., which results in the failure of predictions. Z-number is a more anthropomorphic way of representing uncertain information. The existing references on Z-number research, especially theoretical research, is still in its infancy. A prominent feature of mainstream research in existing theoretical aspects is that the amount of calculation is relatively large, not easy to be understood, and is not conducive to actual engineering applications, particularly inconvenient to handle emergency management that requires high time complexity.

Although the fuzzy analytic hierarchy process overcomes the defects of analytic hierarchy process in the process of calculation, its evaluation results are still calculated based on the experts' scores, which makes the evaluation results inevitably mixed with some experts' personal views. The entropy weight method, by contrast, is mainly based on actual data, without combining some special cases, and the evaluation results are relatively objective. Therefore, I want to obtain the subjective weight and objective weight of each factor through fuzzy analytic hierarchy process and entropy weight method respectively, and then combine the two to obtain their comprehensive weight.

In practical application, the combination of subjective and objective methods are not the same, mainly including mean value method, product method, gray correlation method, etc. However, these combination methods only use the subjective and objective weights of the lowest-level indicators for a relatively simple combination, ignoring the effective integration of the intermediate steps of the two methods. This will cause the calculated weight to be different from the true component in the evaluation process, which deviates from the actual situation. Therefore, a new combination method is adopted, which not only considers the combination of the underlying index weights, but also considers the organic integration of the intermediate processes.

## Definition of construction project risk

In the early stage of decision making, in order to avoid losses, the risks of construction period and cost can be used to evaluate the risk of the whole project before deciding whether to bid. Construction period risk and cost risk can be expressed by Eqs (1) and (2):

$$R_T = \left| \frac{T_\Delta - T_0}{T_\Delta} \right| \tag{1}$$

$$R_C = \left| \frac{C_\Delta - C_0}{C_\Delta} \right| \tag{2}$$

In the formula, $R_T$ and $R_C$ represent the construction period risk and cost risk respectively, $T_\Delta$ and $T_0$ represent the actual construction period and the target construction period respectively, $C_\Delta$ and $C_0$ represent the actual cost and the target cost respectively.

## Fuzzy hierarchy comprehensive weighting method

**Fuzzy consistent judgment matrix.** In order to quantify the decision judgment and form a numerical judgment matrix, the relative importance is given by using the 0.1 ~ 0.9 scale method [23], and the number of index layers is set as $n$, and the initial matrix of Eq (3) is constructed

$$A = (a_{ij})_{n \times n} \tag{3}$$

According to formula (3), the fuzzy consistent discriminant matrix of formula (4) is constructed

$$R = (r_{ij})_{n \times n} \tag{4}$$

In the equation: $\begin{cases} r_{ij} = (r_i - r_j)/2n + 0.5 \\ r_i = \sum\limits_{j=1}^{n} r_{ij}; \ i = 1, 2, \cdots, n \end{cases}$

According to formula (4), the normalized weight of formula (5) is constructed

$$W^R = \left[ \frac{\sum\limits_{j=1}^{n} r_{1j}}{\sum\limits_{i=1}^{n}\sum\limits_{j=1}^{n} r_{ij}}, \frac{\sum\limits_{j=1}^{n} r_{2j}}{\sum\limits_{i=1}^{n}\sum\limits_{j=1}^{n} r_{ij}}, \cdots \frac{\sum\limits_{j=1}^{n} r_{nj}}{\sum\limits_{i=1}^{n}\sum\limits_{j=1}^{n} r_{ij}} \right]^T \tag{5}$$

**Iterative weight of power method.** According to the definition of the power method, the reciprocal matrix of formula (3) is obtained:

$$E = (e_{ij})_{n \times n} \tag{6}$$

In the equation $e_{ij} = r_{ij}/r_{ji}$, $W^R$ is the initial vector $V^{(0)}$, let:

$$V^{(k+1)} = E \frac{V^{(k)}}{\|V^{(k)}\|_{\infty}} \tag{7}$$

In the equation: $\begin{cases} V^{(k)} = (v_1^{(k)}, v_2^{(k)}, \cdots v_n^{(k)}) \\ \|V^{(k)}\|_{\infty} = \max\limits_{1 \le i \le n}(|v_i^{(k)}| \quad ; \text{If } \left| \|V^{(k+1)}\|_{\infty} - \|V^{(k)}\|_{\infty} \right| \le \varepsilon \text{ is satisfied in the} \\ k = 0, 1, 2, \cdots; \end{cases}$

iteration process, the iteration is stopped; wherein $\varepsilon$ is an error, herein take $\varepsilon = 0.0001$. Based on the above derivation, $V^{(k+1)}$ is normalized to obtain the FAHP weight vector:

$$W = \{w_1, w_2, \cdots w_n\} = \{V_1^{(k+1)}/\sum_{i=1}^{n} V_i^{(k+1)}, V_2^{(k+1)}/\sum_{i=1}^{n} V_i^{(k+1)}, \cdots, V_n^{(k+1)}/\sum_{i=1}^{n} V_i^{(k+1)}\}$$

## Entropy weighting method

The data in formula (3) is normalized [24], and the equations are obtained:

$$\begin{cases} R^* = (r_{ij})_{n \times n} \\ 1 \le i \le n; 1 \le j \le n \\ r_{ij} \in [0, 1] \end{cases} \tag{8}$$

define:

$$H_j = -k \sum_{i=1}^{n} d_{ij} \ln d_{ij} \tag{9}$$

Wherein $H_j$ is $j$ th secondary indicators, $d_{ij} = r_{ij}/\sum\limits_{i=1}^{n} r_{ij}$, $k = 1/\ln n$, then the entropy weight of

the $j$th second secondary index is:

$$w_j^* = (1 - H_j)/(n - \sum_{j=1}^{n} H_j) \tag{10}$$

In the equation $0 \leq w_j^* \leq 1$, $\sum_{j=1}^{n} w_j^* = 1$. According to Eq (10), the weights obtained by the entropy weight method can be obtained:

$$W^* = \{w_1^*, w_2^*, \cdots, w_n^*\} \tag{11}$$

Integrate the weights $W$ obtained by FAHP and the weights obtained by entropy weight method, and the EW-FAHP comprehensive weights can be obtained respectively:

$$G = \{g_1, g_2, \cdots, g_n\} \tag{12}$$

In the equation: $g_j = w_j w_j^* / (\sum_{j=1}^{n} w_j w_j^*)$

EW-FAHP combines Fuzzy Analytic Hierarchy Process (FAHP) and Entropy Weight Method (EW), and uses a combination of subjective and objective methods to calculate risk weights. Compared with a single method, the evaluation result is closer to the real situation.

## EW-FAHP weight calculation example analysis

This paper presents the process of determining the weight of each index of public relations risk, and the weights of other risk factors can be determined sequentially.

The relevant data comes from the data of a Sichuan group's entire construction project in a community in Chengdu. First, use the expert scoring method to fill in the proportional scale table for the public relations risk factors of the construction project, and the following matrix can be obtained: $A = \begin{bmatrix} 0.5 & 0.2 & 0.4 & 0.2 & 0.3 \\ 0.8 & 0.5 & 0.7 & 0.6 & 0.7 \\ 0.6 & 0.3 & 0.5 & 0.3 & 0.4 \\ 0.8 & 0.4 & 0.7 & 0.5 & 0.6 \\ 0.7 & 0.3 & 0.6 & 0.4 & 0.5 \end{bmatrix}$

According to formula (4), it can be concluded that:

$R = \begin{bmatrix} 0.5 & 0.33 & 0.45 & 0.36 & 0.41 \\ 0.67 & 0.5 & 0.62 & 0.53 & 0.58 \\ 0.55 & 0.38 & 0.5 & 0.41 & 0.46 \\ 0.64 & 0.47 & 0.59 & 0.5 & 0.55 \\ 0.59 & 0.42 & 0.54 & 0.45 & 0.5 \end{bmatrix}$

From formula (8), we can get: FAHP calculation weight is: $W$ = {0.1341,0.2694,0.1648, 0.2378,0.1939}, from formula (6), the weight of reciprocal matrix is $E$:

$$E = \begin{bmatrix} 1 & 0.4925 & 0.8182 & 0.5625 & 0.6949 \\ 2.0303 & 1 & 1.6316 & 1.1277 & 1.3810 \\ 1.222 & 0.6129 & 1 & 0.6949 & 0.8519 \\ 1.7778 & 0.8868 & 1.4390 & 1 & 1.2222 \\ 1.4390 & 0.7241 & 1.1739 & 0.8182 & 1 \end{bmatrix}$$, from formula (7), the sum row normal-

ized weight initial vector of public risk is obtained: $V = \begin{bmatrix} 0.7069 & 0.4977 \\ 1 & 1 \\ 0.7931 & 0.6117 \\ 0.9483 & 0.8827 \\ 0.8621 & 0.7197 \end{bmatrix}$, from Eq (8), we

can get: $R^* = \begin{bmatrix} 0.1471 & 0.1176 & 0.1379 & 0.1 & 0.12 \\ 0.2533 & 0.2941 & 0.2414 & 0.3 & 0.28 \\ 0.1765 & 0.1765 & 0.1724 & 0.15 & 0.16 \\ 0.2353 & 0.2353 & 0.2414 & 0.25 & 0.24 \\ 0.2059 & 0.1765 & 0.2069 & 0.2 & 0.2 \end{bmatrix}$.

The entropy weight method (EW) calculates the weight as: $W^*$ = {0.809,0.0394,0.0112, 0.0501,0.0258}, based on the above derivation, the EW-FAHP weight of public management risk is G = {0.0066,0.0394,0.0112,0.0501,0.0258}. The FAHP weight and entropy weight (EW) of the remaining secondary indicators can be obtained in turn.

The calculation method of the first-level index weight is the same as that of the second-level index. The initial matrix of first-level indicators is: $A = \begin{bmatrix} 0.5 & 0.7 & 0.6 & 0.6 & 0.8 & 0.9 \\ 0.3 & 0.5 & 0.3 & 0.4 & 0.6 & 0.7 \\ 0.4 & 0.7 & 0.5 & 0.6 & 0.8 & 0.8 \\ 0.4 & 0.6 & 0.4 & 0.5 & 0.7 & 0.8 \\ 0.2 & 0.4 & 0.2 & 0.3 & 0.5 & 0.6 \\ 0.1 & 0.3 & 0.2 & 0.2 & 0.4 & 0.5 \end{bmatrix}$

he comprehensive weight of EW-FAHP is calculated as: G = {0.32299, 0.1126, 0.2568, 0.1878, 0.0745, 0.0443}. Table 1 shows the construction period and cost information of a Sichuan group in a community in Chengdu. According to Eqs (1) and (2), the construction period

**Table 1. Construction information of a residential community in Chengdu, Sichuan.**

| Name of expense | Cost / Yuan | Construction period | Toal area /M² |
|---|---|---|---|
| Early Stage Engineering Cost | ¥98037053.54 | From January 2016 to August 2018, a total of 973 days. | 362528.65 M² |
| Construction and Installation Engineering Costs | ¥738957239.73 | | |
| Basic Engineering Fee | ¥82746945.41 | | |
| Infrastructure Fee | ¥1031000.00 | | |
| Project Management Fee | ¥8799386.87 | | |
| Summation | ¥929571625.55 | | |
| Target Prediction | ¥950000000.00 | 1030 days | |
| Value-at-risk | 0.0215 | 0.0553 | |

risk and cost risk value are obtained. Table 2 shows the weights of relevant indicators at all levels.

To sum up, the risk assessment values of other projects can be obtained by the above methods. For the risk prediction of construction projects, the sample data should reflect its internal laws as much as possible while taking into account its own characteristics. After obtaining the actual conditions of 40 construction projects, 10 experts combined the evaluation indicators to evaluate the risk factors of construction projects, and a total of 40 sets of data were obtained.

In this paper, after obtaining the actual situation of 40 construction projects and combined with evaluation indexes, 10 experts evaluated the risk factors of construction projects, and a total of 40 groups of data were obtained. In view of length, Table 3 gives the comprehensive weights of EW-FAHP for all levels of indicators for 15 groups of construction projects.

## Project risk research model based on convolutional neural network

### The basic principles of convolutional neural networks

With the development of neural networks, convolutional neural networks have been applied in more and more fields. They are currently widely used in visual image analysis, natural language processing and recommendation systems [25], but they have not yet been effectively applied in the risk prediction of construction engineering projects.

**Table 2. Weights of risk factors of construction projects.**

| First Level Index | FAHP Weights | Entropy Weight Method(EW) Weights | EW-FAHP Weights | Secondary Indicators | FAHP Weights | Entropy Weight Method Weights | EW-FAHP |
|---|---|---|---|---|---|---|---|
| $R_1$ | 0.1876 | 0.3010 | 0.3229 | $U_1$ | 0.2132 | 0.3954 | 0.0512 |
| | | | | $U_2$ | 0.1866 | 0.1719 | 0.0195 |
| | | | | $U_3$ | 0.2190 | 0.2580 | 0.0343 |
| | | | | $U_4$ | 0.1817 | 0.0768 | 0.0085 |
| | | | | $U_5$ | 0.1995 | 0.0980 | 0.0119 |
| $R_2$ | 0.1623 | 0.1224 | 0.1136 | $U_6$ | 0.2578 | 0.2622 | 0.0411 |
| | | | | $U_7$ | 0.2756 | 0.4388 | 0.0735 |
| | | | | $U_8$ | 0.2411 | 0.1745 | 0.0256 |
| | | | | $U_9$ | 0.2255 | 0.1245 | 0.0171 |
| $R_3$ | 0.1814 | 0.2475 | 0.2568 | $U_{10}$ | 0.2336 | 0.1380 | 0.0196 |
| | | | | $U_{11}$ | 0.2456 | 0.2549 | 0.0380 |
| | | | | $U_{12}$ | 0.2539 | 0.2949 | 0.0455 |
| | | | | $U_{13}$ | 0.2669 | 0.3123 | 0.0506 |
| $R_4$ | 0.1735 | 0.1892 | 0.1878 | $U_{14}$ | 0.1818 | 0.1066 | 0.0118 |
| | | | | $U_{15}$ | 0.2191 | 0.3756 | 0.0500 |
| | | | | $U_{16}$ | 0.2077 | 0.2245 | 0.0283 |
| | | | | $U_{17}$ | 0.1996 | 0.1439 | 0.0174 |
| | | | | $U_{18}$ | 0.1918 | 0.1494 | 0.0174 |
| $R_5$ | 0.1519 | 0.0858 | 0.0745 | $U_{19}$ | 0.1341 | 0.0809 | 0.0066 |
| | | | | $U_{20}$ | 0.2694 | 0.2409 | 0.0394 |
| | | | | $U_{21}$ | 0.1648 | 0.1120 | 0.0112 |
| | | | | $U_{22}$ | 0.2378 | 0.3472 | 0.0501 |
| | | | | $U_{23}$ | 0.1939 | 0.2190 | 0.0258 |
| $R_6$ | 0.1435 | 0.0540 | 0.0443 | $U_{24}$ | 0.4833 | 0.4008 | 0.1177 |
| | | | | $U_{25}$ | 0.5167 | 0.5992 | 0.1881 |

**Table 3. Risk factor assessment data of construction project.**

| Serial Number | Risk Factor Assessment (Input Unit) | | | | | | Risk (Output Unit) | |
|---|---|---|---|---|---|---|---|---|
| | Economy | Environment | Technology | Social Risk | Public Relations | Natural Risk | Construction Period Risk | Cost Risk |
| 1 | 0.3229 | 0.1136 | 0.2568 | 0.1878 | 0.0745 | 0.043 | 0.0553 | 0.0215 |
| 2 | 0.1307 | 0.0514 | 0.6275 | 0.0291 | 0.1362 | 0.0252 | 0.0809 | 0.0284 |
| 3 | 0.4804 | 0.1130 | 0.1324 | 0.1310 | 0.0793 | 0.0640 | 0.0673 | 0.0266 |
| 4 | 0.1064 | 0.1637 | 0.5275 | 0.0580 | 0.0697 | 0.0746 | 0.0744 | 0.0236 |
| 5 | 0.1964 | 0.0151 | 0.5277 | 0.1477 | 0.0731 | 0.0401 | 0.0753 | 0.0279 |
| 6 | 0.3807 | 0.2807 | 0.1799 | 0.1273 | 0.0157 | 0.0157 | 0.0659 | 0.0242 |
| 7 | 0.2949 | 0.0587 | 0.5505 | 0.0267 | 0.0602 | 0.0091 | 0.0842 | 0.0302 |
| 8 | 0.1006 | 0.1497 | 0.4552 | 0.0843 | 0.1883 | 0.0219 | 0.0682 | 0.0232 |
| 9 | 0.3845 | 0.0286 | 0.2919 | 0.0721 | 0.2132 | 0.0098 | 0.0734 | 0.0278 |
| 10 | 0.2233 | 0.0356 | 0.3026 | 0.1380 | 0.2542 | 0.0465 | 0.0649 | 0.0247 |
| 11 | 0.3641 | 0.0886 | 0.2624 | 0.1515 | 0.0691 | 0.0642 | 0.0685 | 0.0265 |
| 12 | 0.1968 | 0.2046 | 0.3168 | 0.0901 | 0.1871 | 0.0045 | 0.0651 | 0.0234 |
| 13 | 0.1001 | 0.2239 | 0.1025 | 0.0214 | 0.5169 | 0.0353 | 0.0523 | 0.0183 |
| 14 | 0.1926 | 0.1565 | 0.2724 | 0.1827 | 0.1606 | 0.0353 | 0.0610 | 0.0227 |
| 15 | 0.1293 | 0.0833 | 0.5858 | 0.0602 | 0.1035 | 0.0379 | 0.0787 | 0.0244 |

The CNN (Convolutional Neural Networks) network is an extension of DNN (Deep Neural Networks). It mainly includes input layer, output layer, convolutional layer and pooling layer. A convolution kernel of CNN only extracts one feature, and multiple features are extracted by multiple convolution kernels and then integrated in the fully connected layer [11,12]. Fig 3 shows the network structure of 1D-CNN.

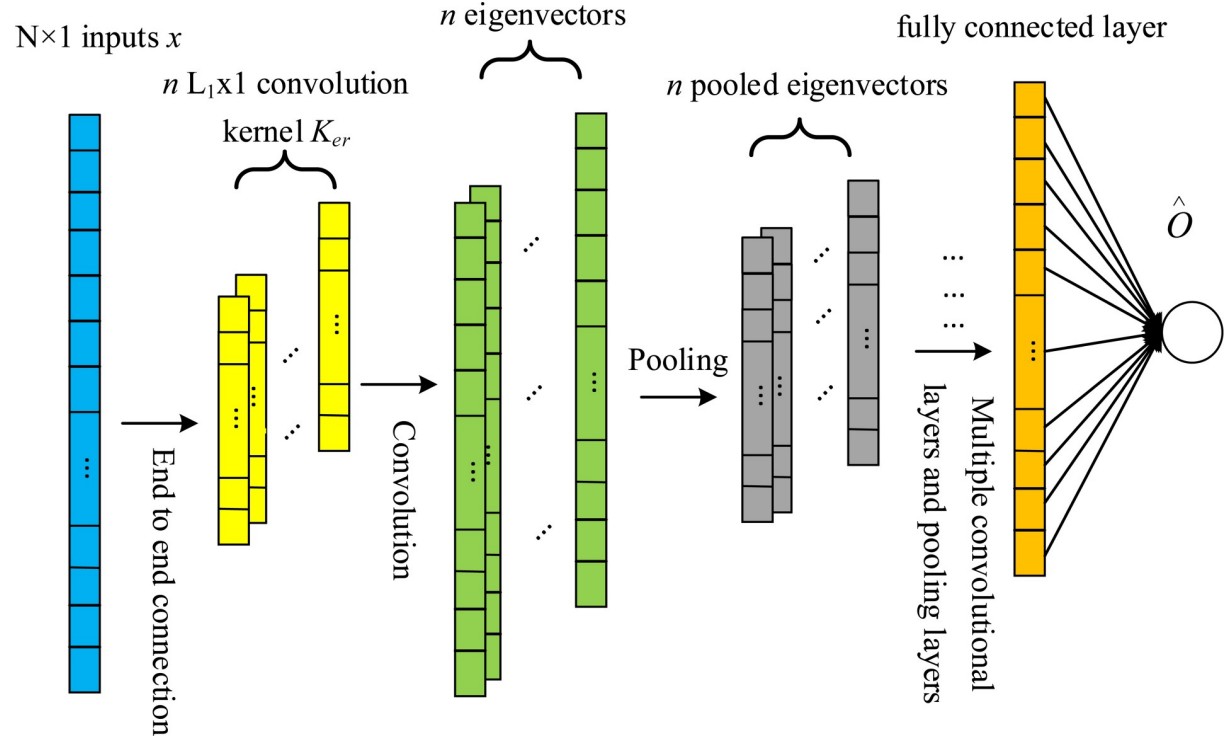

**Fig 3. Network structure of 1D-CNN.**

The 1D-CNN network structure mainly includes five parts: input layer, convolutional layer, pooling layer, fully connected and output layer. For the input one-dimensional information vector, the vector passes through the convolutional layer and the pooling layer. Finally, the corresponding output is obtained through the fully connected layer.

(1) Convolutional layer: Suppose the input signal of the 1D-CNN model is $x$, the length is $N$, and the convolution kernel is used to perform convolution operation on the local area of the input signal. The specific convolution operation formula is:

$$y_i^k = Ker_{L_1}^k * x_i^k + b_i^k \quad (i = 1, 2 \cdots \frac{N}{st_1}) \tag{13}$$

Where: $Ker_{L_1}^k$ represents the kth layer convolution kernel whose length is $L_1$; $*$ indicates that the convolution operation $x_i^k$ means the $i$-th input sub-segment (which is the same length with the convolution kernel); $b_i^k$ represents the offset of the $i$-th convolution output of the k layer; $y_i^k$ represents the convolution output of the $k$-th layer; $st_1$ is the convolution step length, where $y^k = [y_1^k, y_2^k \cdots y_{N/st_1}^k]$.

The non-linear processing of the data after the convolution operation is as follows:

$$s = \max(0, y^k) \tag{14}$$

In the equation above, $s$ represents the activation function of $y^k$. This article uses ReLu, the mainstream activation function in the deep learning world, which can accelerate the model convergence and overcome gradient dispersion.

(2) Pooling layer: Pooling layer reduces the calculation amount and reduces the risk of overfitting by reducing the parameters of the neural network. Maximum pooling can be used to obtain position-independent characteristics. The pooling operation is usually the maximum pooling (max-pooling), as shown in Formula (14), the sequence length can be reduced in dimension.

$$a_j = \max_{(j-1)L_2 \leq t \leq jL_2} (0, s_t^j) \tag{15}$$

In the equation: $j = 1, 2 \cdots \frac{N}{L_2 st_1}$, where $s_t^j$ represents the $t$-th value of the $j$th pooling segment, $a_j$ represents the maximum value of the $j$th pooling segment; $L_2$ represents the length of the pooling segment The output of the pooling layer is:

$$a = [a_1, a_1, \cdots a_{N/L_2 st_1}] \tag{16}$$

Where **a** is the output vector of the pooling layer.

(3) Fully connected layer: The fully connected layer has the same structure as the traditional neural network and is composed of multiple hidden layers. The fully connected layer further abstracts and combines the global timing features, and the output is as follows:

$$\hat{o} = w_o a + b_o \tag{17}$$

In the equation, $w_o$ and $b_o$ are the weight and bias of the fully connected layer, respectively.

## Parameter training of 1D-CNN

Similar to the traditional artificial neural network, the parameter training process of CNN uses the back propagation algorithm [25], and the training process is shown in Fig 4.

1) Initialize the network parameters, construct a network model with appropriate unit depth according to the actual requirements of the construction project risk value samples and

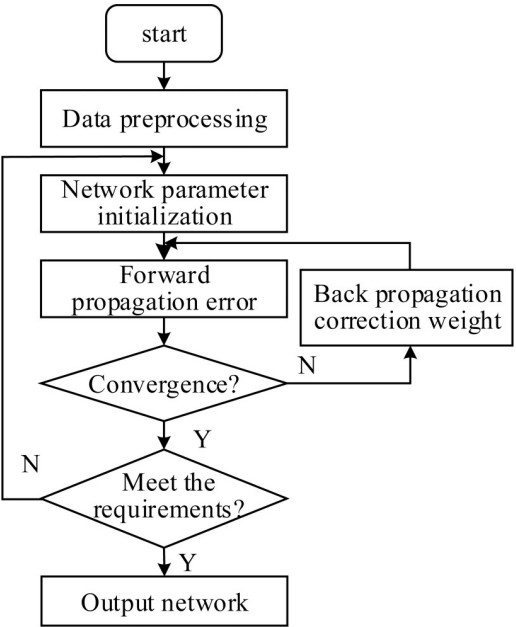

**Fig 4. 1D-CNN parameter training process.**

predictions, and determine the network parameters (such as learning rate, number of iterations, step length, etc.);

2) Input the risk value samples of construction projects into the network, and obtain the error between the network output and the expected target through forward propagation;

3) Determine whether the network converges, if the network does not converge, go to step 4, otherwise go to step 5;

4) Back propagation and weight modification, using BP (Back Propagation) algorithm, the error obtained in step 2 is propagated backward layer by layer to each node, and the weight is updated. Repeat steps 2–4 until the network converges;

5) Determine whether the current network meets the actual requirements according to the recognition accuracy of the test sample, if it meets the requirements, perform step 6, otherwise skip to step 1, and modify the network parameters;

6) Output the average absolute error of the construction project risk and the predicted value of construction period risk and cost risk.

## Simulation analysis

### Operation process of prediction model

Fig 5 is the flow chart of risk prediction model of construction engineering project based on EW-FAHP and 1D-CNN. Firstly, the model identifies the project risk based on the literature analysis method, and then constructs the project risk evaluation index. The relative importance of each risk index is given by using 0.1~0.9 scale method, and the initial matrix is established. Use the initial matrix data to calculate the FAH weights and entropy weights of risk indicators at all levels, and then obtain the comprehensive weight based on EW-FAHP method. The comprehensive weight is input into the 1D-CNN model for learning, and the

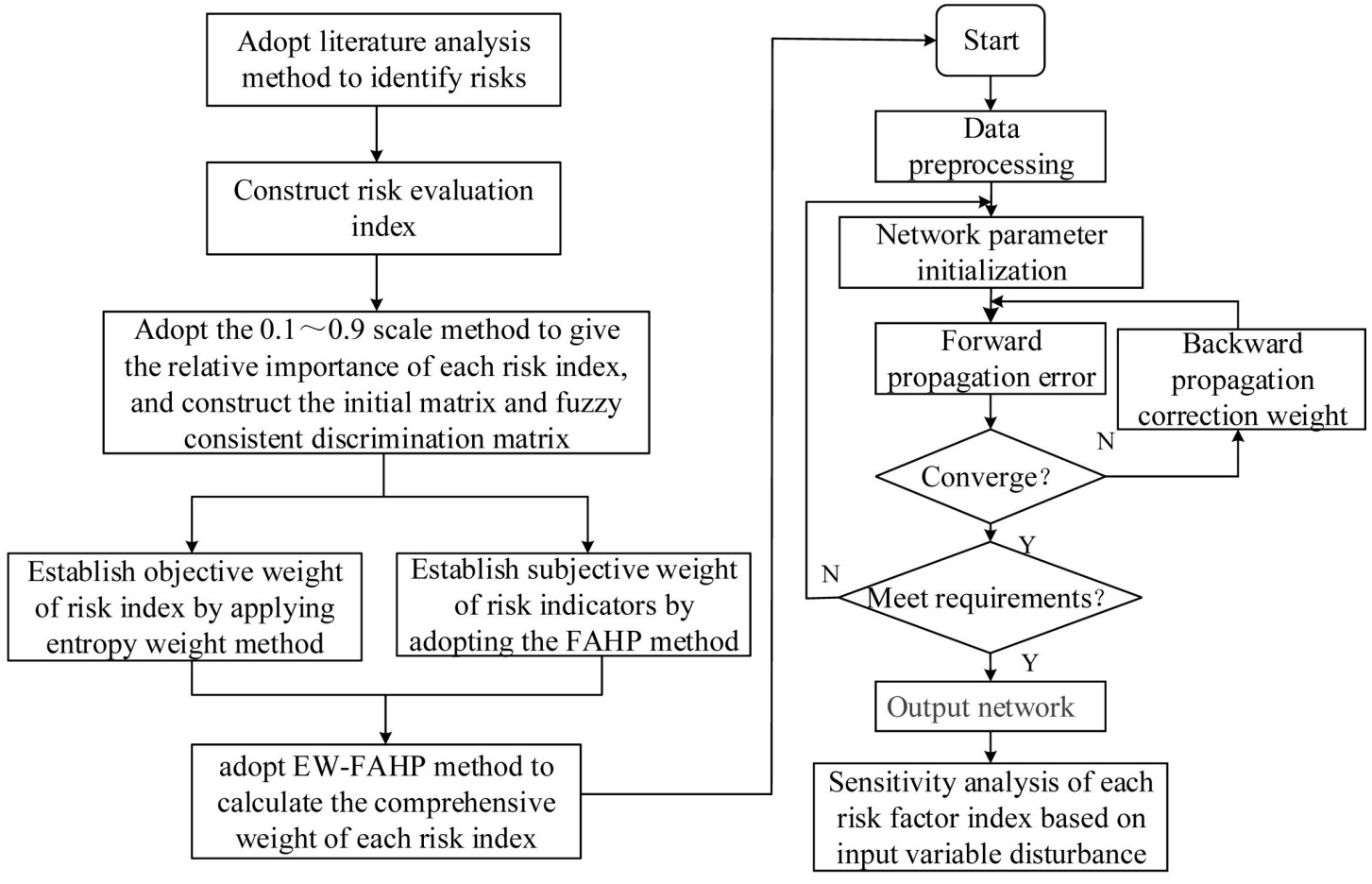

**Fig 5. The flow chart of risk prediction model of construction engineering project based on EW-FAHP and 1D-CNN.**

corresponding model parameters are changed to make the output prediction result converge. Finally, the corresponding prediction results are obtained.

## Simulation conditions

The experimental test platform parameters in this article are Windows 10 Professional 64-bit, processor model (CPU) i7 9850H, main frequency 2.6GHz, and memory (RAM) 2×8GB.

The software framework is the Keras deep learning tool backed by Tensorflow, which can realize rapid model construction and experimental development. In this paper, the number of simulation samples is 40, of which 30 are training samples and 10 are verification samples. Table 4 is the parameter setting of the engineering project risk prediction model based on one-dimensional convolutional neural network (taking cost risk as an example), and similar construction period risks are not listed.

## Analysis of experimental results

Through the prediction of construction project period risk and cost risk, the construction unit can measure the risk of the project and make reasonable decisions based on the predicted value to ensure that the project will not suffer losses.

**Table 4. Parameter settings of project risk prediction model based on one-dimensional convolutional neural network (taking cost risk as an example).**

| Relevant parameters | Type of parameter setting |
|---|---|
| Deep Learning Toolkit | Keras (TensorFlow backend) |
| Nonlinear functions of convolutional layer and fully connected layer | Relu |
| Convolutional layers | 1 |
| Number of fully connected layers | 2 |
| Size of Fully connected layer | 128 and 64 |
| Number of output layers | 1 |
| Size of Output Layer | 1 |
| Size of Convolution Kernel | 128 |
| Length of Convolution Kernel | 1 |
| Index function | MAE |
| Loss function | MSE |
| Optimization function | RMSprop |
| Learning rate | 0.001 |
| Number of iterations | 200 |
| Dropout | 0.4 |

Figs 6 and 7 are the curves of the LOSS values of the training set and the verification set of the construction project duration risk and cost risk calculated by the CNN network with the number of iterations. It can be seen from Figs 6 and 7 that, with the increase of iteration times,

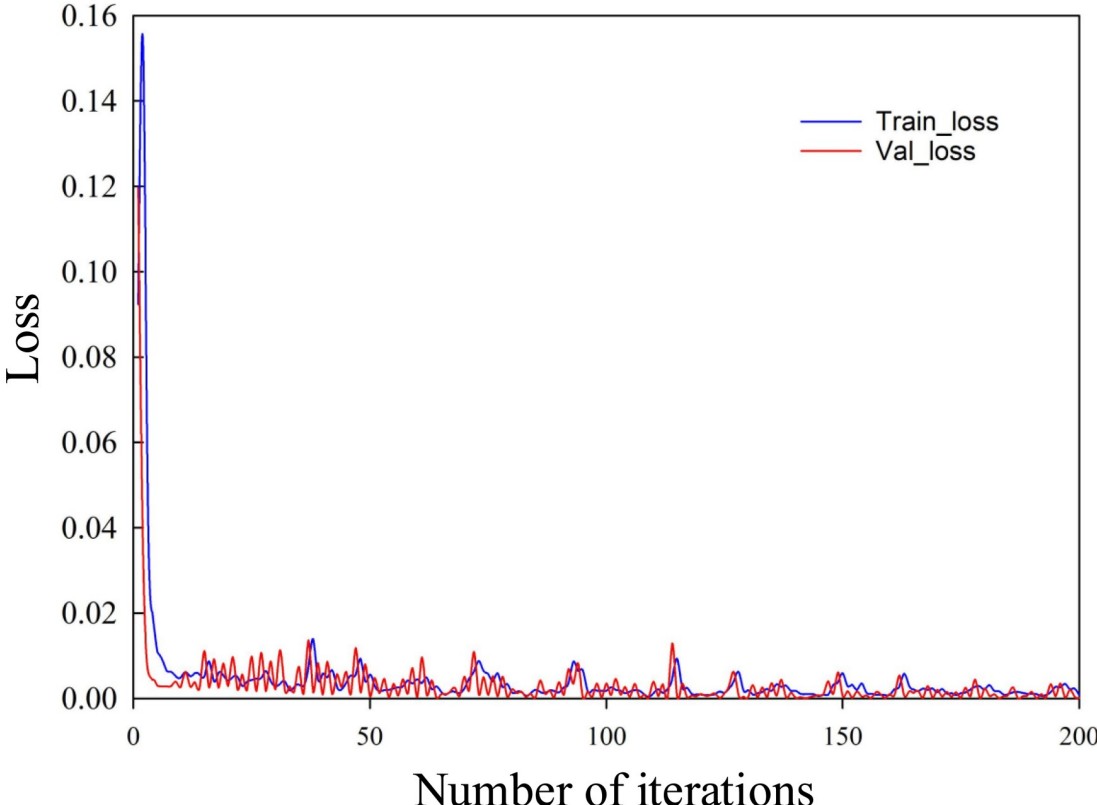

**Fig 6. Curve of value of construction period risk Loss changing with the number of iterations.**

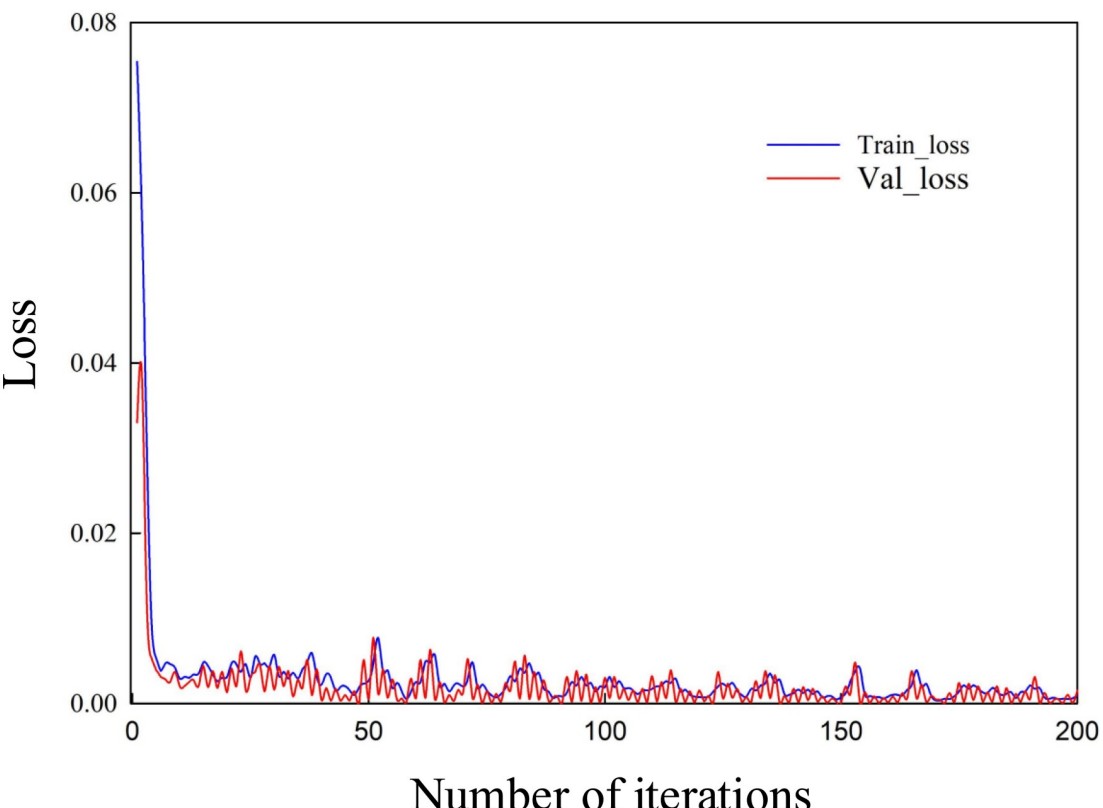

**Fig 7. Curve of cost risk Loss value changing with the number of iterations.**

the LOSS values of training set and verification set keep decreasing and approaching gradually, indicating that the learning process of CNN converges. By performing steps 5 and 6, the predicted values of project duration risk and cost risk can be obtained.

In the experimental results, Mean Absolute errors of construction project duration risk and cost risk are shown in Table 5. The average absolute error index describes the relative deviation degree of risk prediction. The smaller the value is, the higher the prediction accuracy of the model is, and the risk can be effectively predicted.

According to the results in Table 5, the training set MAE of the final results obtained by the CNN prediction model is below 0.002, and the verification set MAE is below 0.001. The proposed algorithm model is convergent, and the model is able to predict risks with high accuracy, meeting the engineering application requirements of the risk prediction of construction projects. Table 6 shows the comparison between the actual and predicted risk values of 10 groups of project No. 6–15 in Table 3.

### Sensitivity analysis based on input variable disturbance

In order to find out the sensitive factors which have an important impact on the construction period and cost indicators from many uncertain factors, and analyze and measure the degree

**Table 5. Mean absolute error value of risk prediction.**

|  | Construction period risk MAE | Cost risk MAE |
|---|---|---|
| **Training set** | 0.00147 | 0.00134 |
| **Validation set** | 0.00044 | 0.00081 |

**Table 6. Relationship between evaluation set and corresponding quantity value.**

| Verification sample number | Output construction period risk | | Output Cost Risk | |
|:---:|:---:|:---:|:---:|:---:|
| | Actual Value | Predicted Value | Actual Value | Predicted Value |
| 1 | 0.0659 | 0.0644 | 0.0242 | 0.0250 |
| 2 | 0.0842 | 0.0844 | 0.0302 | 0.0298 |
| 3 | 0.0682 | 0.0706 | 0.0232 | 0.0243 |
| 4 | 0.0734 | 0.0738 | 0.0278 | 0.0296 |
| 5 | 0.0649 | 0.0650 | 0.0247 | 0.0254 |
| 6 | 0.0685 | 0.0684 | 0.0265 | 0.0269 |
| 7 | 0.0651 | 0.0663 | 0.0234 | 0.0235 |
| 8 | 0.0523 | 0.0524 | 0.0183 | 0.0193 |
| 9 | 0.0610 | 0.0612 | 0.0227 | 0.0224 |
| 10 | 0.0787 | 0.0767 | 0.0244 | 0.0244 |

of influence and sensitivity on the construction period and cost indicators, this article further analyzes the impact of six sensitive factors, including economy, environment, technology, society, public relations, and natural risks, on construction period and cost indicators. For the neural network model, the only things need to know are the input variable data and output data, without the need of prior knowledge. It can carry on training and learning to the training data set, with a lot of simple artificial neuron nonlinear relationship between the simulated data, and can adaptively adjust the connection weight between neurons, so as to establish a network structure that can better reflect the true situation of the data. Therefore, this paper inputs the single sensitive factor into the prediction model under the premise that the other five sensitive factors remain unchanged by +5% and -5%. Sort the sensitivity according to the change size of the output index. Table 7 shows the prediction results when a single sensitive factor changes +5% and -5%.

As can be seen from Table 7, the sensitivity coefficients of the six sensitivity factors are 0.1404, 0.0067, 0.1284, 0.0906, 0.1274 and 0.0636 respectively for the risk of construction period, and 0.0596, 0.0088, 0.0442, 0.0255, 0.0433 and 0.0016 respectively for the risk of cost. Based on the above analysis, it can be seen that for project duration risk and cost risk, the order of sensitivity is economic risk, technical risk, public relations risk, social risk, environmental risk and natural risk.

**Table 7. Evaluation data of construction project risk factors.**

| Risk indicator | Related data | | | | | | Construction period risk | Coefficient of sensitivity | Cost of risk | Coefficient of sensitivity |
|:---:|:---:|:---:|:---:|:---:|:---:|:---:|:---:|:---:|:---:|:---:|
| Economy +5% | 0.3391 | | | | | | 0.0576 | 0.1404 | 0.0238 | 0.0596 |
| Economy -5% | 0.3068 | | | | | | 0.0530 | | 0.0192 | |
| Environment +5% | | 0.1193 | | | | | 0.08094 | 0.0067 | 0.02844 | 0.0088 |
| Environment -5% | | 0.1079 | | | | | 0.08086 | | 0.02836 | |
| Techniques +5% | | | 0.2696 | | | | 0.0689 | 0.1284 | 0.0282 | 0.0442 |
| Techniques -5% | | | 0.2440 | | | | 0.0657 | | 0.0250 | |
| Social risks +5% | | | | 0.1972 | | | 0.0735 | 0.0906 | 0.0227 | 0.0255 |
| Social risks -5% | | | | 0.1784 | | | 0.0753 | | 0.245 | |
| Public relations +5% | | | | | 0.0782 | | 0.0748 | 0.1274 | 0.0274 | 0.0433 |
| Public relations -5% | | | | | 0.0708 | | 0.0758 | | 0.0284 | |
| Natural risk +5% | | | | | | 0.0452 | 0.0660 | 0.0636 | 0.0243 | 0.0016 |
| Natural risk -5% | | | | | | 0.0409 | 0.0658 | | 0.0241 | |

Therefore, comprehensive prediction results and analysis of sensitivity factors show that, for a Sichuan group's construction project in a community in Chengdu, efforts should be made to resolve economic risks, technical risks, and public relations risks before the project starts, so as to avoid project delays and economic losses.

## Comparative analysis of predictive model performance

In order to verify the effect of the 1D-CNN risk prediction model on the risk prediction proposed in this article, BP (back propagation), SVM (Support Vector Machine) and ELM (Extreme Learning Machine) networks were selected to predict and compare the construction project duration risks and cost risks discussed in this article. The sample data uses 10 sets of data of item numbers 6–15 in Table 3 for risk prediction, and compares them with the real values.

Table 8 shows the relevant information of a Sichuan group company in the construction of a residential district in Chengdu, as well as the prediction results by using various prediction models. Fig 8 is a broken line chart of the prediction results of each prediction model, and Fig (a) and Fig (b) in Fig 8 are the comparison charts of prediction results and actual values of duration risk and cost risk by CNN, BP, SVM and ELM respectively.

It can be seen from Table 8 and Fig 8 that the CNN risk prediction curve has the smallest error and the closest curve to the true value, indicating that the CNN risk prediction model proposed in this paper has better prediction accuracy and effect than other risk prediction models.

The prediction model proposed in this paper can be extended in the future research, the prediction results of the risk prediction model proposed in this paper demonstrate that the method has strong practicability in the early stage of the project and in the construction process. Compared with other commonly used prediction algorithms, the prediction accuracy is significantly improved, which has great reference value for project decision-makers.

## Conclusion

This paper proposes a project risk prediction model based on EW-FAHP and one-dimensional convolutional neural network. By selecting the risk evaluation index of construction project, the corresponding risk value is established by combining the EW-FAHP, and then the risk value of construction project is input into the established one-dimensional convolution neural network model for training and learning. The construction project duration risk and cost risk are selected as the output units of the neural network risk prediction. The experimental results show that:

(1) The EW-FAHP weight calculation method proposed in this paper realizes the combination of subjective and objective weighting method and reduces the influence of human factors on the weight. At the same time, the one-dimensional convolutional neural network has strong

**Table 8. Construction information of a residential community in Chengdu, Sichuan.**

| Prediction model | Cost (yuan) | Error | Duration | Error |
|---|---|---|---|---|
| | ¥929571625.55 | | 973 Days | |
| CNN predictive value | ¥930577261.43 | 0.11% | 979 Days | 0.62% |
| BP predictive value | ¥926635809.31 | 0.32% | 988 Days | 1.54% |
| SVM predictive value | ¥951143018.91 | 2.21% | 1015 Days | 4.32% |
| ELM predictive value | ¥970219852.77 | 4.26% | 1051 Days | 8.02% |

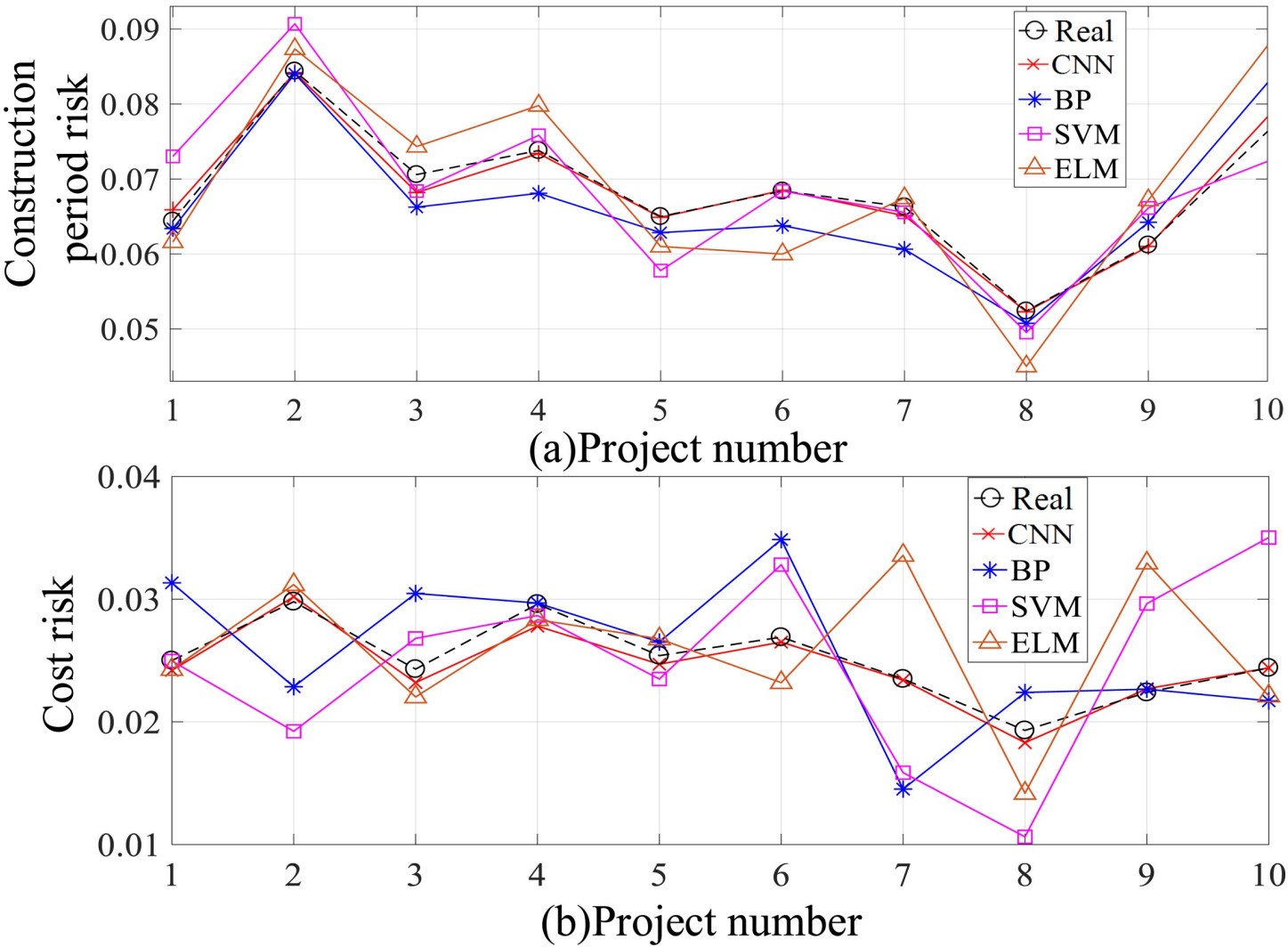

**Fig 8. Comparison diagram of risk predictive values of each prediction model.**

reliability and high accuracy in the prediction of construction project duration risk and cost risk, which can meet the engineering application conditions.

(2) In the case of a certain number of samples, during the neural network training process, the risk Loss value continues to decrease with the number of iterations, and the network converges. This verifies that the risk prediction model has high stability and can provide a reasonable basis for project managers' early decision-making and can effectively reduce risks. It can provide a reasonable basis for project managers' early decision-making and can effectively reduce risks. In addition, due to the difficulty of obtaining sample data, if more relevant data of construction projects can be obtained by combining relevant big data, the prediction accuracy of the 1D-CNN risk prediction model will be further improved, and the prediction results will be more convincing.

In the practical engineering application, it aims at different types of risk prediction requirements, such as investment risk, traffic risk, coal mine safety and disease risks, etc. After sorting and collecting relevant data, EW-FAHP or other combination of subjective and objective weight determination methods are applied to determine the comprehensive weight of risks

affecting the prediction results. Then, the comprehensive weight data is input into the 1D-CNN prediction model for learning and prediction, and the prediction results are also of great reference significance.

## Author Contributions

**Data curation:** Leilei Chen.

**Funding acquisition:** Yawen Zhong.

**Methodology:** Yawen Zhong.

**Supervision:** Hailing Li.

**Writing – original draft:** Yawen Zhong.

**Writing – review & editing:** Yawen Zhong.

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
