## [Decision Letter · Decision Letter 0]

23 Nov 2020

PONE-D-20-30147

Construction Project Risk Prediction Model Based on EW-FAHP and One Dimensional Convolution Neural Network

PLOS ONE

Dear Dr. zhong,

Thank you for submitting your manuscript to PLOS ONE. After careful consideration, we feel that it has merit but does not fully meet PLOS ONE’s publication criteria as it currently stands. Therefore, we invite you to submit a revised version of the manuscript that addresses the points raised during the review process.

We look forward to receiving your revised manuscript.

Kind regards,

Dragan Pamucar

Academic Editor

PLOS ONE

Journal Requirements:

2.Thank you for stating the following in the Acknowledgments Section of your manuscript:

[This research was supported by National Natural Science Foundation of China (Nos. 11705122, 61640223),

Sichuan Provincial Department of Science and Technology Project (No. 2019YJ0477), Artificial intelligence Sichuan Key Laboratory Project (No.2019RYY01), Nanchong Science and Technology Bureau Project (No.19SXHZ0040)]

 [no]

Reviewers' comments:

Reviewer's Responses to Questions

**Comments to the Author**

1. Is the manuscript technically sound, and do the data support the conclusions?

Reviewer #1: No

Reviewer #2: Yes

2. Has the statistical analysis been performed appropriately and rigorously? 

Reviewer #1: No

Reviewer #2: Yes

3. Have the authors made all data underlying the findings in their manuscript fully available?

Reviewer #1: No

Reviewer #2: Yes

4. Is the manuscript presented in an intelligible fashion and written in standard English?

Reviewer #1: Yes

Reviewer #2: Yes

5. Review Comments to the Author

Reviewer #1: The author(s) need to consider the following points as limitation or further scope for refining the paper:

- Introduction should be clearly stated research questions and targets first. Then answer several questions: Why is the topic important (or why do you study on it)? What are the research questions? What are your contributions? Why is to propose this particular methods? The last two questions are answered in some parts in the Introduction section. But, the answer is not presented in a proper way. You should provide more information in this regard.

- Need to highlight the novelty of study in the introduction.

- I suggest authors to clearly summarize what specific advantages brings your approach. Enrich your Introduction section with more explanation: Why do you present this approach? Why you use Entropy method for criteria weighting and not the other objective methods like CRITIC or FANMA method? Why do we FAHP since can use only Entropy method for determining criteria weights?

- Literature review should be revised. Remove lumped references. All references cited in the text should be explained and discussed in the text. Remove some old references published before 2017-2018. Also, literature review should be presented in a better way. You should discuss application of various MCDM tools different fields. You should update your literature review with a papers published in last two-three years, and remove old references. I suggest authors to read and cite below interesting references: Petrovic, I., & Kankaras, M. (2020). A hybridized IT2FS-DEMATEL-AHP-TOPSIS multicriteria decision making approach: Case study of selection and evaluation of criteria for determination of air traffic control radar position. Decision Making: Applications in Management and Engineering, 3(1), 146-164.;

Badi, I., Abdulshahed, A., Shetwan, A., & Eltayeb, W. (2019). Evaluation of solid waste treatment methods in Libya by using the analytic hierarchy process. Decision Making: Applications in Management and Engineering, 2(2), 19-35.

Stanković, M., Gladović, P., & Popović, V. (2019). Determining the importance of the criteria of traffic accessibility using fuzzy AHP and rough AHP method. Decision Making: Applications in Management and Engineering, 2(1), 86-104.

- Add flowchart of proposed methodology and follow that flowchart steps in case study.

- Case study should be better organized. The calculations should be deeply presented and follow the methodology presented in methodology section. Add more deep calculations in case study section.

- Add sensitivity analysis and validation of the results.

- The problem on which this present method is applied has significant social and managerial implications. How the method can address those implications need to be included.

- Conclusion- Add future scope. Also, how the proposed method can be applicable to other real life problems need to be mentioned. Add limitations of proposed model. Do not use bullets or numerations in this section.

Reviewer #2: Thank you for inviting me as a reviewer for manuscript titled: Construction Project Risk Prediction Model Based on EW-FAHP and One Dimensional Convolution Neural Network. The presented methodology has great potential. Although the paper is well written, in my opinion the paper would be more exiting if you implement below improvements:

- Indicate what is new that is presented in the paper.

- Show the entire model in phases and steps in one figure.

- One unit should be a description of the FAHP method. Analyze different approaches in fuzzyfication of AHP methods (standard fuzzy numbers, interval fuzzy numbers, Z numbers ...)

- The paper lacks sensitivity analysis.

- Reference analysis is not at a satisfactory level. Most of the reference are of older date.

6. PLOS authors have the option to publish the peer review history of their article (what does this mean?). If published, this will include your full peer review and any attached files.

Reviewer #1: No

Reviewer #2: No

---

## [Author Response · Author response to Decision Letter 0]

7 Jan 2021

Dear editor：

Hello! The article " Construction Project Risk Prediction Model Based on EW-FAHP and One Dimensional Convolution Neural Network" has been revised in strict accordance with the opinions of the review experts. The specific changes are as follows, please review.

Reviewer #1: 

Comments on Revision1：

- Introduction should be clearly stated research questions and targets first. Then answer several questions: Why is the topic important (or why do you study on it)? What are the research questions? What are your contributions? Why is to propose this particular methods? The last two questions are answered in some parts in the Introduction section. But, the answer is not presented in a proper way. You should provide more information in this regard.

- Need to highlight the novelty of study in the introduction.

- I suggest authors to clearly summarize what specific advantages brings your approach. Enrich your Introduction section with more explanation: Why do you present this approach? Why you use Entropy method for criteria weighting and not the other objective methods like CRITIC or FANMA method? Why do we FAHP since can use only Entropy method for determining criteria weights?

Modification instructions ：

Introduction

With the continuous development of science and technology, the complexity of construction projects continues to increase, the construction period continues to grow, and there are many uncertain factors[1], in order to reduce the probability of risk occurrence and effectively avoid potential risks to the entire project, it is necessary to predict the risks of the construction projects. In reference [2], the subway project construction risk management method is based on Bayesian network. In reference[3], the Fault Tree Analysis (FTA) method is combined with Bayesian network, and a Bayesian network-based shale gas well blowout risk analysis method is proposed. However, Bayesian networks are based on prior probabilities. In many cases, prior probabilities depend on assumptions, which will largely lead to poor prediction results. Combine with AHP theory, use rough set to analyze project risk group decision to realize attribute reduction, and combine with analytic hierarchy process to realize quantitative research and analysis of project risk. Literature [4] uses AHP to evaluate solid waste treatment methods in Libya, but it is difficult for AHP to check and adjust the consistency of the judgment matrix. Literature [5] proposed the use of fuzzy analytic hierarchy process and rough analytic hierarchy process to evaluate traffic accessibility method. Literature[6] proposed the use of IT2FS-DEMATEL to eliminate less important indicators, combines the IT2FS-AHP method to sort the final indicators, and establishes a multi-index decision-making model. But literature [5] and literature [6] involve risk prediction, rough set and IT2FS-DEMATEL may have a greater impact on the final prediction results after removing redundant attributes.

With the development of artificial intelligence and big data, neural network has attracted more and more researchers’ attention. Because neural network has a strong nonlinear fitting ability and has a good effect on mapping nonlinear relations, relevant scholars have combined neural network with engineering project risk research in recent years and achieved certain results [6][7].Literature [8] proposed a railway construction risk assessment algorithm based on BP neural network. The expert scoring method was used to establish initial sample data, and the BP neural network prediction model was used to learn and predict the samples to get the risk score of each construction project. However, BP neural network has some problems, such as not the best approximation of continuous function and long training time. Literature [9] proposed a project risk evaluation algorithm based on PCA (principal component analysis)-RBF neural network on the basis of BP model, which improved the shortcomings of BP neural network that it is difficult to obtain the optimal network, but the RBF neural network the center of the hidden basis function is selected in the input sample set, which in many cases can hardly reflect the real input-output relationship of the system. In order to solve the problems of the above-mentioned neural network, literature [10] proposed a method for predicting the risk of underground engineering rockburst based on ANN and ABC. (the artificial neural network (ANN) and artificial bee colony (ABC), in order to further improve the prediction accuracy, the paper uses the artificial bee colony algorithm to optimize the artificial bee colony algorithm, but the artificial bee colony algorithm has weak search ability and relatively slow convergence speed. 

As construction projects become large-scale and risk factors continue to increase, traditional risk predictions mostly use regular event analysis, correlation analysis and other methods to analyze key indicators and detailed records, which rely heavily on manual extraction by professional workers. The current risk assessment of construction projects adopts a single expert scoring method, entropy weight method, analytic hierarchy process or fuzzy analytic hierarchy process, which does not fully combine multiple evaluation methods, resulting in incomplete de-tailed factors affecting project risks, and lack of objectivity and accuracy of inspection and evaluation results. At present, Analytic Hierarchy Process (AHP) and its derivative methods are still the most widely used and most effective risk assessment in the complex large systems. Among them, the Fuzzy Analytic Hierarchy Process (FAHP), which integrates fuzzy theories, improves the weight determination problem of AHP, and its practicality and simplicity have been applied more and more widely [11]. In order to improve the closeness between the weight of evaluation index and reality, this paper adopts the entropy weight-fuzzy analytic hierarchy process (EW-FAHP method) to determine the weight. The risk prediction method based on traditional neural network risk prediction requires too many weights, which reduces the accuracy of project prediction to a certain extent [12]. The current risk assessment of construction projects uses a single CNN network with different convolution kernels to perform convolution operations on the input data, thereby obtaining global features of the data, and then down-sampling the extracted features through pooling operations, reducing the amount of calculations. At the same time, it can also suppress the overfitting of the model to a certain extent.

Therefore, this paper uses entropy weight method and fuzzy analytic hierarchy process to evaluate the construction period and cost index system of the construction project, proposes a construction project risk prediction model based on EW-FAHP and 1D-CNN, identifies the existing risks of the construction project through reference analytical method and constructs risk evaluation index system. Using Entropy Weight (EW) and Fuzzy Analytic Hierarchy Process (FAHP), the risk weight of each risk evaluation index is determined by combining subjective and objective evaluation methods. One dimensional convolution neural network model is constructed to train and learn the risk weight of construction project. The duration risk and cost risk of construction project are selected as the output unit of convolution neural network. The aver-age absolute error between the predicted value and the actual value of duration risk and cost risk is analyzed to realize the risk prediction of construction project.

The prediction results of the risk prediction model proposed in this paper show that the method has strong practicability in the early stage of the project and in the construction process. Compared with other commonly used forecasting algorithms, the forecasting accuracy has been significantly improved, which is of greater reference value for project decision-makers.

Comments on Revision 2

- Literature review should be revised. Remove lumped references. All references cited in the text should be explained and discussed in the text. Remove some old references published before 2017-2018. Also, literature review should be presented in a better way. You should discuss application of various MCDM tools different fields. You should update your literature review with a papers published in last two-three years, and remove old references. I suggest authors to read and cite below interesting references: Petrovic, I., & Kankaras, M. (2020). A hybridized IT2FS-DEMATEL-AHP-TOPSIS multicriteria decision making approach: Case study of selection and evaluation of criteria for determination of air traffic control radar position. Decision Making: Applications in Management and Engineering, 3(1), 146-164.;

Badi, I., Abdulshahed, A., Shetwan, A., & Eltayeb, W. (2019). Evaluation of solid waste treatment methods in Libya by using the analytic hierarchy process. Decision Making: Applications in Management and Engineering, 2(2), 19-35.

Stanković, M., Gladović, P., & Popović, V. (2019). Determining the importance of the criteria of traffic accessibility using fuzzy AHP and rough AHP method. Decision Making: Applications in Management and Engineering, 2(1), 86-104. 

Modification instructions ：

Thanks for the comments of the review experts, I have put the relevant references into the article for discussion, namely, reference [4][5][6]

In reference [2], the subway project construction risk management method is based on Bayesian network. In reference [3], the fault tree analysis(FTA) method is combined with Bayesian network, and the shale gas well blowout risk analysis method based on Bayesian network is proposed. However, Bayesian network is based on prior probability, which often depends on assumptions, which leads to poor prediction effect to a large extent. Literature [4] uses AHP to evaluate solid waste treatment methods in Libya, but it is difficult for AHP to check and adjust the consistency of the judgment matrix. Literature [5] proposed the use of fuzzy analytic hierarchy process and rough analytic hierarchy process to evaluate traffic accessibility method, and literature [6] proposed the use of IT2FS-DEMATEL. The method eliminates less important indicators, combines the IT2FS-AHP method to sort the final indicators, and establishes a multi-index decision-making model. However, in terms of reference [5] and reference [6] concerning risk prediction, rough set and IT2FS-DEMATEL may have a great influence on the final prediction results after eliminating redundant attributes.

Comments on Revision 3

- Add flowchart of proposed methodology and follow that flowchart steps in case study.

Modification instructions ：

Fig. 5 is the flow chart of risk prediction model of construction engineering project based on EW-FAHP and 1D-CNN. Firstly, the model identifies the project risk based on the literature analysis method, and then constructs the project risk evaluation index. The relative importance of each risk index is given by using 0.1~0.9 scale method, and the initial matrix is established. Use the initial matrix data to calculate the FAH weights and entropy weights of risk indicators at all levels, and then obtain the comprehensive weight based on EW-FAHP method. The comprehensive weight is input into the 1D-CNN model for learning, and the corresponding model parameters are changed to make the output prediction result converge. Finally, the corresponding prediction results are obtained.

Fig.5. The flow chart of risk prediction model of construction engineering project based on EW-FAHP and 1D-CNN

Comments on Revision 4

- Case study should be better organized. The calculations should be deeply presented and follow the methodology presented in methodology section. Add more deep calculations in case study section.

Modification instructions：

This paper presents the process of determining the weight of each index of public relations risk, and the weights of other risk factors can be determined sequentially.

The relevant data comes from the data of a Sichuan group's entire construction project in a community in Chengdu. First, use the expert scoring method to fill in the proportional scale table for the public relations risk factors of the construction project, and the following matrix can be obtained: 

According to formula (4), it can be concluded that: 

 From formula (8), we can get: FAHP calculation weight is: W={0.1341,0.2694,0.1648,0.2378,0.1939}, from formula (6), the weight of reciprocal matrix is : , from formula (7), the sum row normalized weight initial vector of public risk is obtained: , from equation (8), we can get: .

The entropy weight method (EW) calculates the weight as: W*={0.809,0.0394,0.0112,0.0501,0.0258}, based on the above derivation, the EW-FAHP weight of public management risk is G={0.0066,0.0394,0.0112,0.0501,0.0258}. The FAHP weight and entropy weight (EW) of the remaining secondary indicators can be obtained in turn.

The calculation method of the first-level index weight is the same as that of the second-level index. The initial matrix of first-level indicators is： 

The comprehensive weight of EW-FAHP is calculated as: G={0.32299, 0.1126, 0.2568, 0.1878, 0.0745, 0.0443}. Table 1 shows the construction period and cost information of a Sichuan group in a community in Chengdu. According to equation (1) and equation (2), the construction period risk and cost risk value are obtained. Table 2 shows the weights of relevant indicators at all levels.

Comments on Revision 4

- Add sensitivity analysis and validation of the results.

Modification instructions：

Sensitivity analysis based on input variable disturbance

In order to find out the sensitive factors which have an important impact on the construction period and cost indicators from many uncertain factors, and analyze and measure the degree of influence and sensitivity on the construction period and cost indicators, this article further analyzes the impact of six sensitive factors, including economy, environment, technology, society, public relations, and natural risks, on construction period and cost indicators. For the neural network model, the only things need to know are the input variable data and output data, without the need of prior knowledge. It can carry on training and learning to the training data set, with a lot of simple artificial neuron nonlinear relationship between the simulated data, and can adaptively adjust the connection weight between neurons, so as to establish a network structure that can better reflect the true situation of the data. Therefore, this paper inputs the single sensitive factor into the prediction model under the premise that the other five sensitive factors remain unchanged by +5% and -5%. Sort the sensitivity according to the change size of the output index. Table 7 shows the prediction results when a single sensitive factor changes +5% and -5%. 

Table 7 Evaluation data of construction project risk factors

Risk indicator Related data Construction period

risk Coefficient of sensitivity

 Cost of risk Coefficient of sensitivity

Economy

+5% 0.3391 0.0576 0.1404 0.0238 0.0596

Economy

-5% 0.3068 0.0530 0.0192 

Environment

+5% 0.1193 0.08094 0.0067 0.02844 0.0088

Environment -5% 0.1079 0.08086 0.02836 

Techniques

+5% 0.2696 0.0689 0.1284 0.0282 0.0442

Techniques

-5% 0.2440 0.0657 0.0250 

Social risks

+5% 0.1972 0.0735 0.0906 0.0227 0.0255

Social risks

-5% 0.1784 0.0753 0.245 

Public relations

+5% 0.0782 0.0748 0.1274 0.0274 0.0433

Public relations

-5% 0.0708 0.0758 0.0284 

Natural risk

+5% 0.0452 0.0660 0.0636 0.0243 0.0016

Natural risk

-5% 0.0409 0.0658 0.0241 

As can be seen from Table 7, the sensitivity coefficients of the six sensitivity factors are 0.1404, 0.0067, 0.1284, 0.0906, 0.1274 and 0.0636 respectively for the risk of construction period, and 0.0596, 0.0088, 0.0442, 0.0255, 0.0433 and 0.0016 respectively for the risk of cost. Based on the above analysis, it can be seen that for project duration risk and cost risk, the order of sensitivity is economic risk, technical risk, public relations risk, social risk, environmental risk and natural risk.

Therefore, comprehensive prediction results and analysis of sensitivity factors show that, for a Sichuan group's construction project in a community in Chengdu, efforts should be made to resolve economic risks, technical risks, and public relations risks before the project starts, so as to avoid project delays and economic losses.

Comments on Revision 5

- The problem on which this present method is applied has significant social and managerial implications. How the method can address those implications need to be included.

- Conclusion- Add future scope. Also, how the proposed method can be applicable to other real life problems need to be mentioned. Add limitations of proposed model. Do not use bullets or numerations in this section.

Modification instructions：

The prediction model proposed in this paper can be extended in the future research, the prediction results of the risk prediction model proposed in this paper demonstrate that the method has strong practicability in the early stage of the project and in the construction process. Compared with other commonly used prediction algorithms, the prediction accuracy is significantly improved, which has great reference value for project decision-makers.

Conclusion

This paper proposes a project risk prediction model based on EW-FAHP and one-dimensional convolutional neural network. By selecting the risk evaluation index of construction project, the corresponding risk value is established by combining the EW-FAHP, and then the risk value of construction project is input into the established one-dimensional convolution neural network model for training and learning. The construction project duration risk and cost risk are selected as the output units of the neural network risk prediction. The experimental results show that: 

(1) The EW-FAHP weight calculation method proposed in this paper realizes the combination of subjective and objective weighting method and reduces the influence of human factors on the weight. At the same time, the one-dimensional convolutional neural network has strong reliability and high accuracy in the pre-diction of construction project duration risk and cost risk, which can meet the engineering application conditions.

 (2) In the case of a certain number of samples, during the neural network training process, the risk Loss value continues to decrease with the number of iterations, and the network converges. This verifies that the risk prediction model has high stability and can provide a reasonable basis for project managers’ early decision-making and can effectively reduce risks. It can provide a reasonable basis for project managers’ early decision-making and can effectively reduce risks. In addition, due to the difficulty of obtaining sample data, if more relevant data of construction projects can be obtained by combining relevant big data, the prediction accuracy of the 1D-CNN risk prediction model will be further improved, and the prediction results will be more convincing.

In the practical engineering application, it aims at different types of risk prediction requirements, such as investment risk, traffic risk, coal mine safety and disease risks, etc. After sorting and collecting relevant data, EW-FAHP or other combination of subjective and objective weight determination methods are applied to determine the comprehensive weight of risks affecting the prediction results. Then, the comprehensive weight data is input into the 1D-CNN prediction model for learning and prediction, and the prediction results are also of great reference significance.

Reviewer #2: 

Comments on Revision1：

- Indicate what is new that is presented in the paper.

Modification instructions：

Abstract

 In order to solve the problem of low accuracy of traditional construction project risk prediction, a project risk prediction model based on EW-FAHP and 1D-CNN(One Dimensional Convolution Neural Network) is proposed. Firstly, the risk evaluation index value of construction project is selected by literature analysis method, and the comprehensive weight of risk index is obtained by combining entropy weight method (EW) and fuzzy analytic hierarchy process (FAHP). The risk weight is input into the 1D-CNN model for training and learning, and the pre-diction values of construction period risk and cost risk are output to realize the risk prediction. The experimental results show that the average absolute error of the construction period risk and cost risk of the risk prediction model proposed in this paper is below 0.1%, which can meet the risk prediction of construction projects with high accuracy.

Introduction

With the continuous development of science and technology, the complexity of construction projects continues to increase, the construction period continues to grow, and there are many uncertain factors[1], in order to reduce the probability of risk occurrence and effectively avoid potential risks to the entire project, it is necessary to predict the risks of the construction projects. 

In reference [2], the subway project construction risk management method is based on Bayesian network. In reference[3], the Fault Tree Analysis (FTA) method is combined with Bayesian network, and a Bayesian network-based shale gas well blowout risk analysis method is proposed. However, Bayesian networks are based on prior probabilities. In many cases, prior probabilities depend on assumptions, which will largely lead to poor prediction results. Combine with AHP theory, use rough set to analyze project risk group decision to realize attribute reduction, and combine with analytic hierarchy process to realize quantitative research and analysis of project risk. Literature [4] uses AHP to evaluate solid waste treatment methods in Libya, but it is difficult for AHP to check and adjust the consistency of the judgment matrix. Literature [5] proposed the use of fuzzy analytic hierarchy process and rough analytic hierarchy process to evaluate traffic accessibility method. Literature[6] proposed the use of IT2FS-DEMATEL to eliminate less important indicators, combines the IT2FS-AHP method to sort the final indicators, and establishes a multi-index decision-making model. But literature [5] and literature [6] involve risk prediction, rough set and IT2FS-DEMATEL may have a greater impact on the final prediction results after removing redundant attributes.

With the development of artificial intelligence and big data, neural network has attracted more and more researchers’ attention. Because neural network has a strong nonlinear fitting ability and has a good effect on mapping nonlinear relations, relevant scholars have combined neural network with engineering project risk research in recent years and achieved certain results [6][7]. Literature [8] proposed a railway construction risk assessment algorithm based on BP neural network. The expert scoring method was used to establish initial sample data, and the BP neural network prediction model was used to learn and predict the samples to get the risk score of each construction project. However, BP neural network has some problems, such as not the best approximation of continuous function and long training time. Literature [9] proposed a project risk evaluation algorithm based on PCA (principal component analysis)-RBF neural network on the basis of BP model, which improved the shortcomings of BP neural network that it is difficult to obtain the optimal network, but the RBF neural network the center of the hidden basis function is selected in the input sample set, which in many cases can hardly reflect the real input-output relationship of the system. In order to solve the problems of the above-mentioned neural network, literature [10] proposed a method for predicting the risk of underground engineering rockburst based on ANN and ABC. (the artificial neural network (ANN) and artificial bee colony (ABC), in order to further improve the prediction accuracy, the paper uses the artificial bee colony algorithm to optimize the artificial bee colony algorithm, but the artificial bee colony algorithm has weak search ability and relatively slow convergence speed.As construction projects become large-scale and risk factors continue to increase, traditional risk predictions mostly use regular event analysis, correlation analysis and other methods to analyze key indicators and detailed records, which rely heavily on manual extraction by professional workers. The current risk assessment of construction projects adopts a single expert scoring method, entropy weight method, analytic hierarchy process or fuzzy analytic hierarchy process, which does not fully combine multiple evaluation methods, resulting in incomplete de-tailed factors affecting project risks, and lack of objectivity and accuracy of inspection and evaluation results. At present, Analytic Hierarchy Process (AHP) and its derivative methods are still the most widely used and most effective risk assessment in the complex large systems. Among them, the Fuzzy Analytic Hierarchy Process (FAHP), which integrates fuzzy theories, improves the weight determination problem of AHP, and its practicality and simplicity have been applied more and more widely [11]. In order to improve the closeness between the weight of evaluation index and reality, this paper adopts the entropy weight-fuzzy analytic hierarchy process (EW-FAHP method) to determine the weight. The risk prediction method based on traditional neural network risk prediction requires too many weights, which reduces the accuracy of project prediction to a certain extent [12]. The current risk assessment of construction projects uses a single CNN network with different convolution kernels to perform convolution operations on the input data, thereby obtaining global features of the data, and then down-sampling the extracted features through pooling operations, reducing the amount of calculations. At the same time, it can also suppress the overfitting of the model to a certain extent.

Therefore, this paper uses entropy weight method and fuzzy analytic hierarchy process to evaluate the construction period and cost index system of the construction project, proposes a construction project risk prediction model based on EW-FAHP and 1D-CNN, identifies the existing risks of the construction project through reference analytical method and constructs risk evaluation index system. Using Entropy Weight (EW) and Fuzzy Analytic Hierarchy Process (FAHP), the risk weight of each risk evaluation index is determined by combining subjective and objective evaluation methods. One dimensional convolution neural network model is constructed to train and learn the risk weight of construction project. The duration risk and cost risk of construction project are selected as the output unit of convolution neural network. The aver-age absolute error between the predicted value and the actual value of duration risk and cost risk is analyzed to realize the risk prediction of construction project.

The prediction results of the risk prediction model proposed in this paper show that the method has strong practicability in the early stage of the project and in the construction process. Compared with other commonly used forecasting algorithms, the forecasting accuracy has been significantly improved, which is of greater reference value for project decision-makers.

Comments on Revision2：

- Show the entire model in phases and steps in one figure.

Modification instructions：

Fig. 5 is the flow chart of risk prediction model of construction engineering project based on EW-FAHP and 1D-CNN. Firstly, the model identifies the project risk based on the literature analysis method, and then constructs the project risk evaluation index. The relative importance of each risk index is given by using 0.1~0.9 scale method, and the initial matrix is established. Use the initial matrix data to calculate the FAH weights and entropy weights of risk indicators at all levels, and then obtain the comprehensive weight based on EW-FAHP method. The comprehensive weight is input into the 1D-CNN model for learning, and the corresponding model parameters are changed to make the output prediction result converge. Finally, the corresponding prediction results are obtained.

Fig.5. The flow chart of risk prediction model of construction engineering project based on EW-FAHP and 1D-CNN

Comments on Revision3：

- One unit should be a description of the FAHP method. Analyze different approaches in fuzzyfication of AHP methods (standard fuzzy numbers, interval fuzzy numbers, Z numbers ...)

Modification instructions：

Through the research and analysis of interval fuzzy numbers, it is found that the existing processing method is to directly model and predict the two boundary points. Doing so often leads to a failure to describe the overall development trend of the sequence and the results predicted by the model are prone to be confused, etc., which results in the failure of predictions. Z-number is a more anthropomorphic way of representing un-certain information. The existing references on Z-number research, especially theoretical research, is still in its infancy. A prominent feature of mainstream research in existing theoretical aspects is that the amount of calculation is relatively large, not easy to be understood, and is not conducive to actual engineering applications, particularly inconvenient to handle emergency management that requires high time complexity.

Although the fuzzy analytic hierarchy process overcomes the defects of analytic hierarchy process in the process of calculation, its evaluation results are still calculated based on the experts' scores, which makes the evaluation results inevitably mixed with some experts' personal views. The entropy weight method, by contrast, is mainly based on actual data, without combining some special cases, and the evaluation results are relatively objective. Therefore, I want to obtain the subjective weight and objective weight of each factor through fuzzy analytic hierarchy process and entropy weight method respectively, and then combine the two to obtain their comprehensive weight. 

In practical application, the combination of subjective and objective methods are not the same, mainly including mean value method, product method, gray correlation method, etc. However, these combination methods only use the subjective and objective weights of the lowest-level indicators for a relatively simple combination, ignoring the effective integration of the intermediate steps of the two methods. This will cause the calculated weight to be different from the true component in the evaluation process, which deviates from the actual situation. Therefore, a new combination method is adopted, which not only considers the combination of the underlying index weights, but also considers the organic integration of the intermediate processes.

Comments on Revision4：

- The paper lacks sensitivity analysis.

Modification instructions：

Sensitivity analysis based on input variable disturbance

In order to find out the sensitive factors which have an important impact on the construction period and cost indicators from many uncertain factors, and analyze and measure the degree of influence and sensitivity on the construction period and cost indicators, this article further analyzes the impact of six sensitive factors, including economy, environment, technology, society, public relations, and natural risks, on construction period and cost indicators. For the neural network model, the only things need to know are the input variable data and output data, without the need of prior knowledge. It can carry on training and learning to the training data set, with a lot of simple artificial neuron nonlinear relationship between the simulated data, and can adaptively adjust the connection weight between neurons, so as to establish a network structure that can better reflect the true situation of the data. Therefore, this paper inputs the single sensitive factor into the prediction model under the premise that the other five sensitive factors remain unchanged by +5% and -5%. Sort the sensitivity according to the change size of the output index. Table 7 shows the prediction results when a single sensitive factor changes +5% and -5%. 

Table 7 Evaluation data of construction project risk factors

Risk indicator Related data Construction period

risk Coefficient of sensitivity

 Cost of risk Coefficient of sensitivity

Economy

+5% 0.3391 0.0576 0.1404 0.0238 0.0596

Economy

-5% 0.3068 0.0530 0.0192 

Environment

+5% 0.1193 0.08094 0.0067 0.02844 0.0088

Environment -5% 0.1079 0.08086 0.02836 

Techniques

+5% 0.2696 0.0689 0.1284 0.0282 0.0442

Techniques

-5% 0.2440 0.0657 0.0250 

Social risks

+5% 0.1972 0.0735 0.0906 0.0227 0.0255

Social risks

-5% 0.1784 0.0753 0.245 

Public relations

+5% 0.0782 0.0748 0.1274 0.0274 0.0433

Public relations

-5% 0.0708 0.0758 0.0284 

Natural risk

+5% 0.0452 0.0660 0.0636 0.0243 0.0016

Natural risk

-5% 0.0409 0.0658 0.0241 

As can be seen from Table 7, the sensitivity coefficients of the six sensitivity factors are 0.1404, 0.0067, 0.1284, 0.0906, 0.1274 and 0.0636 respectively for the risk of construction period, and 0.0596, 0.0088, 0.0442, 0.0255, 0.0433 and 0.0016 respectively for the risk of cost. Based on the above analysis, it can be seen that for project duration risk and cost risk, the order of sensitivity is economic risk, technical risk, public relations risk, social risk, environmental risk and natural risk.

Therefore, comprehensive prediction results and analysis of sensitivity factors show that, for a Sichuan group's construction project in a community in Chengdu, efforts should be made to resolve economic risks, technical risks, and public relations risks before the project starts, so as to avoid project delays and economic losses.

Comments on Revision4：

- Reference analysis is not at a satisfactory level. Most of the reference are of older date.

Modification instructions：

The references in this paper have been updated.

References

1. Wang QK, Wang YH. ANP- based Research on the Strategic Risk Assessment for Multi Project Management in Prefabricated Buildings [J]. JOURNAL OF WUHAN UNIVERSITY OF TECHNOLOGY, 2018,40(4):76-79

2. Xiao QD, Zhao ZN, Liu LC. Research on Construction Risk Management of Subway Project Based on Bayesian Network [J]. Journal of Xinyang Normal University(Natural Science Edition) https://kns.cnki.net/kcms/detail/41.1107.N.20201207.1007.002.html.

3. Chen K, Chen X，Wei X et al. Bayesian network-based risk analysis on the b low out of the shale gas wells [J]. Journal of Safety and Environment，2019,19(6):226-241.

4. Stanković, M., Gladović, P., & Popović, V. Determining the importance of the criteria of traffic accessibility using fuzzy AHP and rough AHP method[J]. Decision Making: Applications in Management and Engineering, 2019,2(1): 86-104

5. Badi, I., Abdulshahed, A., Shetwan, A., & Eltayeb, W. Evaluation of solid waste treatment methods in Libya by using the analytic hierarchy process[J]. Decision Making: Applications in Management and Engineering, 2019,2(2):19-35.

6. Petrovic, I., & Kankaras, M. A hybridized IT2FS-DEMATEL-AHP-TOPSIS multicriteria decision making approach: Case study of selection and evaluation of criteria for determination of air traffic control radar position[J]. Decision Making: Applications in Management and Engineering,2020, 3(1), 146-164.

7. Ehsan E, Nima K, Ezutah U O, el at. Applying fuzzy multi-objective linear programming to a project management decision with nonlinear fuzzy membership functions[J]. Neural Computing and Applications,2017,(28)8:2193-2206.

8. Jin J, Li ZH , Zhu L, et al. Application of BP Neural Network in Risk Evaluation of Railway Construction[J]. JOURNAL OF RAILWAY ENGINEERING SOCIETY 2019,3:103-109

9. Lu XQ, Huang YJ, Wang X. Intelligent Evaluation Model Based on PCA-RBF Neural Network Applied to Risk Assessment of PPP Projects[J].2017,14:59-63.

10. Zhou, J., Koopialipoor, M., Li, E. et al. Prediction of rockburst risk in underground projects developing a neuro-bee intelligent system[J]. Bull Eng Geol Environ 2020,79, 4265–4279.

11. Gao, Cl., Li, Sc., Wang, J. et al. The Risk Assessment of Tunnels Based on Grey Correlation and Entropy Weight Method[J]. Geotech Geol Eng, 2018,36, 1621–1631.

12. Khorram, S. Correction to: A novel approach for ports’ container terminals’ risk management based on formal safety assessment: FAHP-entropy measure—VIKOR model[J]. Nat Hazards 103, 1709 (2020).

13. Guo YH, Shi YC, Xu YJ. Evaluation of Bridge Construction Quality based on Improved FAHP Evaluation Method[J]. Journal of Civil Engineering and Management, 2017,34(1):44-48.

14. Zhou FY, Jin LP, Dong Jun. A Review of Convolutional Neural Networks [J].Journal of Computers，2017,40(6):1229-1251.

15. Morgunova E.P. Investment Project Risk Identification and Evaluation[C]. In: Solovev D. (eds) Smart Technologies and Innovations in Design for Control of Technological Processes and Objects: Economy and Production. FarEastСon 2018. Smart Innovation, Systems and Technologies, vol 138. Springer, Cham. 

16. Sanghera P. Project Risk Management. In: CAPM® in Depth[M]. 2019, Apress, Berkeley, CA. 

17. Schatteman D, Herroelen W, STIJN V D V, et al. Methodology for integrated risk management and proactive scheduling of construction projects[J]. Faculty of Economics and Applied Economics, doi:10.2139/ssrn.950903

18. Mulgan, G. Artificial intelligence and collective intelligence: the emergence of a new field [J]. AI & Soc 2018,33, 631–632. 

19. Anysz, H., Buczkowski, B. The association analysis for risk evaluation of significant delay occurrence in the completion date of construction project [J]. Int. J. Environ. Sci. Technol. 2019,16, 5369–5374.

20. Yu XJ, Peng YY. The Application and Challenges of Artificial Intelligence in the Field of Financial Risk Management[J]. Southern Finance, 2017,9:70-74.

21. Wu Q, Gao SH, Zhou T. Comprehensive Evaluation of Construction Project Schedule Control [J]. Journal of Xi'an University of Science and Technology, 2011, 4(31):412-419. 

22. Li, S.C., Wu, J. A multi-factor comprehensive risk assessment method of karst tunnels and its engineering application [J]. Bull Eng Geol Environ 2019,78, 1761–1776. 

23. Zhong YW. Study on Schedule Risk of Project Group based on Rough Set Theory [D]. Chengdu: Xihua University，2018.

24. Li L, Li SY, He WJ, et al. Emergency Capability Evaluation of Construction Projects based on EM and FAHP [J]. Journal of Xi'an University of Science and Technology, 2020,4(40):572-579.

25. Yu C, Luo B, Wang DG, et al. Evaluation of Cultivated Land Consolidation Potential Based on Improved FAHP-Entropy Weighting Method[J]. China Agricultural Resources and Regional Planning, 2020,41(6):15-24.

26. Liu W, Dong WQ. Research on Risk Assessment Method of Drainage Pipe Network Based on AHP-Entropy Method Combination Weighting[J].Journal of Safety and Environment, https://doi.org/10.13637/j.issn.1009-6094.2019.1400

27. Niu XX, Suen C Y. A Novel Hybrid CNN–SVM Classifier for Recognizing Handwritten Digits[J]. Pattern Recognition, 2012,45(4):1318–1325.

---

## [Decision Letter · Decision Letter 1]

21 Jan 2021

Construction Project Risk Prediction Model Based on EW-FAHP and One Dimensional Convolution Neural Network

PONE-D-20-30147R1

Dear Dr. zhong,

We’re pleased to inform you that your manuscript has been judged scientifically suitable for publication and will be formally accepted for publication once it meets all outstanding technical requirements.

Kind regards,

Dragan Pamucar

Academic Editor

PLOS ONE

Additional Editor Comments (optional):

Reviewers' comments:

Reviewer's Responses to Questions

**Comments to the Author**

1. If the authors have adequately addressed your comments raised in a previous round of review and you feel that this manuscript is now acceptable for publication, you may indicate that here to bypass the “Comments to the Author” section, enter your conflict of interest statement in the “Confidential to Editor” section, and submit your "Accept" recommendation.

Reviewer #1: (No Response)

Reviewer #2: All comments have been addressed

2. Is the manuscript technically sound, and do the data support the conclusions?

Reviewer #1: (No Response)

Reviewer #2: Yes

3. Has the statistical analysis been performed appropriately and rigorously? 

Reviewer #1: (No Response)

Reviewer #2: Yes

4. Have the authors made all data underlying the findings in their manuscript fully available?

Reviewer #1: (No Response)

Reviewer #2: Yes

5. Is the manuscript presented in an intelligible fashion and written in standard English?

Reviewer #1: (No Response)

Reviewer #2: Yes

6. Review Comments to the Author

Reviewer #1: (No Response)

Reviewer #2: All the reviewers' comments have been addressed carefully and sufficiently, the revisions are rational from my point of view, I think the current version of the paper can be accepted.

7. PLOS authors have the option to publish the peer review history of their article (what does this mean?). If published, this will include your full peer review and any attached files.

Reviewer #1: No

Reviewer #2: No

---

## [Editor Report · Acceptance letter]

25 Jan 2021

PONE-D-20-30147R1 

Construction Project Risk Prediction Model Based on EW-FAHP and One Dimensional Convolution Neural Network 

Dear Dr. Zhong:

I'm pleased to inform you that your manuscript has been deemed suitable for publication in PLOS ONE. Congratulations! Your manuscript is now with our production department. 

Kind regards, 

on behalf of

Dr. Dragan Pamucar 

Academic Editor

PLOS ONE